# Vision GNN: An Image is Worth Graph of Nodes

**Kai Han**[1,2*]   **Yunhe Wang**[2*]   **Jianyuan Guo**[2]   **Yehui Tang**[2,3]   **Enhua Wu**[1,4]

[1]State Key Lab of Computer Science, ISCAS & UCAS
[2]Huawei Noah's Ark Lab
[3]Peking University   [4]University of Macau
{kai.han,yunhe.wang}@huawei.com, weh@ios.ac.cn

## Abstract

Network architecture plays a key role in the deep learning-based computer vision system. The widely-used convolutional neural network and transformer treat the image as a grid or sequence structure, which is not flexible to capture irregular and complex objects. In this paper, we propose to represent the image as a graph structure and introduce a new *Vision GNN* (ViG) architecture to extract graph-level feature for visual tasks. We first split the image to a number of patches which are viewed as nodes, and construct a graph by connecting the nearest neighbors. Based on the graph representation of images, we build our ViG model to transform and exchange information among all the nodes. ViG consists of two basic modules: Grapher module with graph convolution for aggregating and updating graph information, and FFN module with two linear layers for node feature transformation. Both isotropic and pyramid architectures of ViG are built with different model sizes. Extensive experiments on image recognition and object detection tasks demonstrate the superiority of our ViG architecture. We hope this pioneering study of GNN on general visual tasks will provide useful inspiration and experience for future research.

The PyTorch code is available at https://github.com/huawei-noah/Efficient-AI-Backbones and the MindSpore code is available at https://gitee.com/mindspore/models.

## 1   Introduction

In the modern computer vision system, convolutional neural networks (CNNs) used to be the de-facto standard network architecture [29, 27, 17]. Recently, transformer with attention mechanism was introduced for visual tasks [9, 3] and attained competitive performance. MLP-based (multi-layer perceptron) vision models [49, 50] can also work well without using convolutions or self-attention. These progresses are pushing the vision models towards an unprecedented height.

Different networks treat the input image in different ways. As shown in Figure 1, the image data is usually represented as a regular grid of pixels in the Euclidean space. CNNs [29] apply sliding window on the image and introduce the shift-invariance and locality. The recent vision transformer [9] or MLP [49] treats the image as a sequence of patches. For example, ViT [9] divides a $224 \times 224$ image into a number of $16 \times 16$ patches and forms a sequence with length of 196 as input.

Instead of the regular grid or sequence representation, we process the image in a more flexible way. One basic task of computer vision is to recognize the objects in an image. Since the objects are usually not quadrate whose shape is irregular, the commonly-used grid or sequence structures in previous networks like ResNet and ViT are redundant and inflexible to process them. An object can be viewed as a composition of parts, *e.g.*, a human can be roughly divided into head, upper body,

---

[*]Equal contribution.

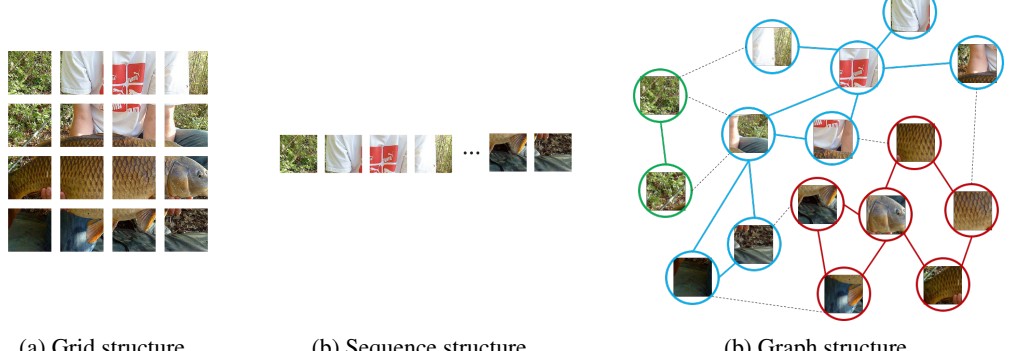

(a) Grid structure.      (b) Sequence structure.      (b) Graph structure.

Figure 1: Illustration of the grid, sequence and graph representation of the image. In the grid structure, the pixels or patches are ordered only by the spatial position. In the sequence structure, the 2D image is transformed to a sequence of patches. In the graph structure, the nodes are linked by its content and are not constrained by the local position.

arms and legs. These parts linked by joints naturally form a graph structure. By analyzing the graph, we are able to recognize the human. Moreover, graph is a generalized data structure that grid and sequence can be viewed as a special case of graph. Viewing an image as a graph is more flexible and effective for visual perception.

Based on the graph representation of images, we build the vision graph neural network (ViG for short) for visual tasks. Instead of treating each pixel as a node which will result in too many nodes (>10K), we divide the input image to a number of patches and view each patch as a node. After constructing the graph of image patches, we use our ViG model to transform and exchange information among all the nodes. The basic cells of ViG include two parts: Grapher and FFN (feed-forward network) modules. Grapher module is constructed based on graph convolution for graph information processing. To alleviate over-smoothing phenomenon of conventional GNN, a FFN module is utilized for node feature transformation and encouraging node diversity. With Grapher and FFN modules, we build our ViG models in both isotropic and pyramid manners. In the experiments, we demonstrate the effectiveness of ViG model on visual tasks like image classification and object detection. For instance, our Pyramid ViG-S achieves 82.1% top-1 accuracy on ImageNet classification task, which outperforms the representative CNN (ResNet [17]), MLP (CycleMLP [5]) and transformer (Swin-T [35]) with similar FLOPs (about 4.5G). To the best of our knowledge, our work is the first to successfully apply graph neural network on large-scale visual tasks. We hope our work will inspire the community to further explore more powerful network architectures.

## 2 Related Work

In this section, we first revisit the backbone networks in computer vision. Then we review the development of graph neural network, especially GCN and its applications on visual tasks.

### 2.1 CNN, Transformer and MLP for Vision

The mainstream network architecture in computer vision used to be convolutional network [29, 27, 17]. Starting from LeNet [29], CNNs have been successfully used in various visual tasks, *e.g.*, image classification [27], object detection [42] and semantic segmentation [36]. The CNN architecture is evolving rapidly in the last ten years. The representative works include ResNet [17], MobileNet [21] and NAS [75, 70]. Vision transformer was introduced for visual tasks from 2020 [14, 9, 3, 4]. From then on, a number of variants of ViT [9] were proposed to improve the performance on visual tasks. The main improvements include pyramid architecture [57, 35], local attention [15, 35] and position encoding [61]. Inspired by vision transformer, MLP is also explored in computer vision [49, 50]. With specially designed modules [5, 32, 12, 48], MLP can achieve competitive performance and work on general visual tasks like object detection and segmentation.

## 2.2 Graph Neural Network

The earliest graph neural network was initially outlined in [11, 44]. Micheli [38] proposed the early form of spatial-based graph convolutional network by architecturally composite nonrecursive layers. In recent several years, the variants of spatial-based GCNs have been introduced, such as [39, 1, 10]. Spectral-based GCN was first presented by Bruna *et al.* [2] that introduced graph convolution based on the spectral graph theory. Since this time, a number of works to improve and extend spectral-based GCN have been proposed [18, 7, 26]. The GCNs are usually applied on graph data, such as social networks [13], citation networks [45] and biochemical graphs [55].

The applications of GCN in the field of computer vision [63, 28, 56, 25] mainly include point clouds classification, scene graph generation, and action recognition. A point cloud is a set of 3D points in space which is usually collected by LiDAR scans. GCN has been explored for classifying and segmenting points clouds [28, 58, 69]. Scene graph generation aims to parse the input image intro a graph with the objects and their relationship, which is usually solved by combining object detector and GCN [63, 68]. By processing the naturally formed graph of linked human joints, GCN was utilized on human action recognition task [24, 67]. GCN can only tackle specific visual tasks with naturally constructed graph. For general applications in computer vision, we need a GCN-based backbone network that directly processes the image data.

## 3 Approach

In this section, we describe how to transform an image to a graph and introduce vision GNN architectures to learn visual representation.

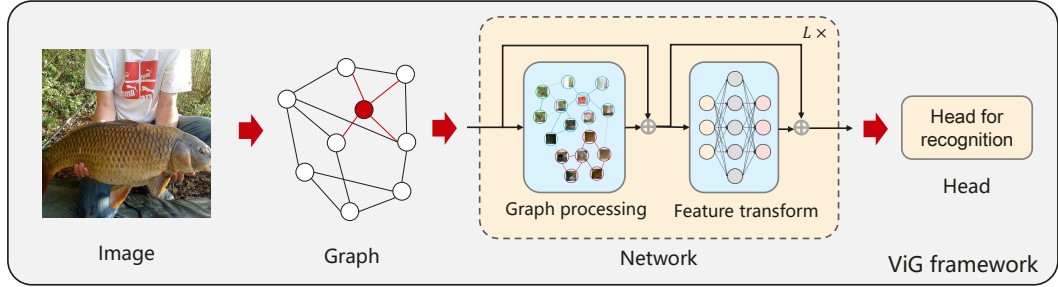

Figure 2: The framework of the proposed ViG model.

### 3.1 ViG Block

**Graph Structure of Image.**    For an image with size of $H \times W \times 3$, we divided it into $N$ patches. By transforming each patch into a feature vector $\mathbf{x}_i \in \mathbb{R}^D$, we have $X = [\mathbf{x}_1, \mathbf{x}_2, \cdots, \mathbf{x}_N]$ where $D$ is the feature dimension and $i = 1, 2, \cdots, N$. These features can be viewed as a set of unordered nodes which are denoted as $\mathcal{V} = \{v_1, v_2, \cdots, v_N\}$. For each node $v_i$, we find its $K$ nearest neighbors $\mathcal{N}(v_i)$ and add an edge $e_{ji}$ directed from $v_j$ to $v_i$ for all $v_j \in \mathcal{N}(v_i)$. Then we obtain a graph $\mathcal{G} = (\mathcal{V}, \mathcal{E})$ where $\mathcal{E}$ denote all the edges. We denote the graph construction process as $\mathcal{G} = G(X)$ in the following. By viewing the image as a graph data, we explore how to utilize GNN to extract its representation.

The advantages of graph representation of the image include: 1) graph is a generalized data structure that grid and sequence can be viewed as a special case of graph; 2) graph is more flexible than grid or sequence to model the complex object as an object in the image is usually not quadrate whose shape is irregular; 3) an object can be viewed as a composition of parts (*e.g.*, a human can be roughly divided into head, upper body, arms and legs), and graph structure can construct the connections among those parts; 4) the advanced research on GNN can be transferred to address visual tasks.

**Graph-level processing.**    To be general, we start from the features $X \in \mathbb{R}^{N \times D}$. We first construct a graph based on the features: $\mathcal{G} = G(X)$. A graph convolutional layer can exchange information between nodes by aggregating features from its neighbor nodes. Specifically, graph convolution

operates as follows:

$$\mathcal{G}' = F(\mathcal{G}, \mathcal{W})$$
$$= Update(Aggregate(\mathcal{G}, W_{agg}), W_{update}), \qquad (1)$$

where $W_{agg}$ and $W_{update}$ are the learnable weights of the aggregation and update operations, respectively. More concretely, the aggregation operation computes the representation of a node by aggregating features of neighbor nodes, and the update operation further merge the aggregated feature:

$$\mathbf{x}'_i = h(\mathbf{x}_i, g(\mathbf{x}_i, \mathcal{N}(\mathbf{x}_i), W_{agg}), W_{update}), \qquad (2)$$

where $\mathcal{N}(\mathbf{x}_i^l)$ is the set of neighbor nodes of $\mathbf{x}_i^l$. Here we adopt max-relative graph convolution [30] for its simplicity and efficiency:

$$g(\cdot) = \mathbf{x}''_i = [\mathbf{x}_i, \max(\{\mathbf{x}_j - \mathbf{x}_i | j \in \mathcal{N}(\mathbf{x}_i)\})], \qquad (3)$$
$$h(\cdot) = \mathbf{x}'_i = \mathbf{x}''_i W_{update}, \qquad (4)$$

where the bias term is omitted. The above graph-level processing can be denoted as $X' = \text{GraphConv}(X)$.

We further introduce multi-head update operation of graph convolution. The aggregated feature $\mathbf{x}''_i$ is first split into $h$ heads, *i.e.*, $head^1, head^2, \cdots, head^h$ and then these heads are updated with different weights respectively. All the heads can be updated in parallel and are concatenated as the final values:

$$\mathbf{x}'_i = [head^1 W_{update}^1, head^2 W_{update}^2, \cdots, head^h W_{update}^h]. \qquad (5)$$

Multi-head update operation allows the model to update information in multiple representation subspaces, which is beneficial to the feature diversity.

**ViG block.** The previous GCNs usually repeatedly use several graph convolution layers to extract aggregated feature of the graph data. The over-smoothing phenomenon in deep GCNs [31, 40] will decrease the distinctiveness of node features and lead to performance degradation for visual recognition, as shown in Figure 3 where diversity is measured as $\|X - \mathbf{1}\tilde{\mathbf{x}}^T\|$ with $\tilde{\mathbf{x}} = \arg\min_{\tilde{\mathbf{x}}} \|X - \mathbf{1}\tilde{\mathbf{x}}^T\|$ [8]. To alleviate this issue, we introduce more feature transformations and nonlinear activations in our ViG block.

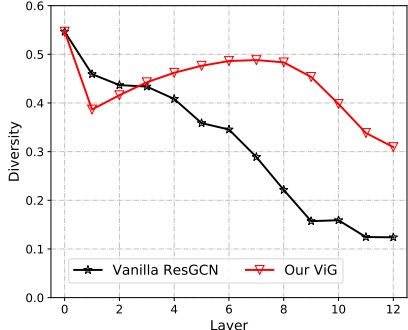

Figure 3: Feature diversity of nodes as layer changes.

We apply a linear layer before and after the graph convolution to project the node features into the same domain and increase the feature diversity. A nonlinear activation function is inserted after graph convolution to avoid layer collapse. We call the upgraded module as Grapher module. In practice, given the input feature $X \in \mathbb{R}^{N \times D}$, the Grapher module can be expressed as

$$Y = \sigma(\text{GraphConv}(X W_{in})) W_{out} + X, \qquad (6)$$

where $Y \in \mathbb{R}^{N \times D}$, $W_{in}$ and $W_{out}$ are the weights of fully-connected layers, $\sigma$ is the activation function, *e.g.*, ReLU and GeLU [19], and the bias term is omitted.

To further encourage the feature transformation capacity and relief the over-smoothing phenomenon, we utilize feed-forward network (FFN) on each node. The FFN module is a simple multi-layer perceptron with two fully-connected layers:

$$Z = \sigma(Y W_1) W_2 + Y, \qquad (7)$$

where $Z \in \mathbb{R}^{N \times D}$, $W_1$ and $W_2$ are the weights of fully-connected layers, and the bias term is omitted. The hidden dimension of FFN is usually larger than $D$. In both Grapher and FFN modules, batch normalization is applied after every fully-connected layer or graph convolution layer, which is omitted in Eq. 6 and 7 for concision. A stack of Grapher module and FFN module constitutes the ViG block which serves as the basic building unit for constructing a network. Based on the graph representation of images and the proposed ViG block, we can build the ViG network for visual tasks as shown in Figure 2. Compared to vanilla ResGCN [30], our ViG can maintain the feature diversity (Figure 3) as the layer goes deeper so as to learn discriminative representations.

## 3.2 Network Architecture

In the field of computer vision, the commonly-used transformer usually has an isotropic architecture (*e.g.*, ViT [9]), while CNNs prefer to use pyramid architecture (*i.e.*, ResNet [17]). To have a extensive comparison with other types of neural networks, we build two kinds of network architectures for ViG, *i.e.*, isotropic architecture and pyramid architecture.

**Isotropic architecture.** Isotropic architecture means the main body has features with equal size and shape throughout the network, such as ViT [9] and ResMLP [50]. We build three versions of isotropic ViG architecture with different models sizes, *i.e.*, ViG-Ti, S and B. The number of nodes is set as $N = 196$. To enlarge the receptive field gradually, the number of neighbor nodes $K$ increases from 9 to 18 linearly as the layer goes deep in these three models. The number of heads is set as $h = 4$ by default. The details are listed in Table 1.

Table 1: Variants of our isotropic ViG architecture. The FLOPs are calculated for the image with 224×224 resolution. 'Ti' denotes tiny, 'S' denotes small, and 'B' denotes base.

| Model | Depth | Dimension $D$ | Params (M) | FLOPs (B) |
|-------|-------|---------------|------------|-----------|
| ViG-Ti | 12 | 192 | 7.1 | 1.3 |
| ViG-S | 16 | 320 | 22.7 | 4.5 |
| ViG-B | 16 | 640 | 86.8 | 17.7 |

**Pyramid architecture.** Pyramid architecture considers the multi-scale property of images by extracting features with gradually smaller spatial size as the layer goes deeper, such as ResNet [17] and PVT [57]. Empirical evidences show that pyramid architecture is effective for visual tasks [57]. Thus, we utilize the advanced design and build four versions of pyramid ViG models. The details are shown in Table 2. Note that we utilize the spatial reduction [57] in the first two stages to handle large number of nodes.

Table 2: Detailed settings of Pyramid ViG series. $D$: feature dimension, $E$: hidden dimension ratio in FFN, $K$: number of neighbors in GCN, $H \times W$: input image size. 'Ti' denotes tiny, 'S' denotes small, 'M' denotes medium, and 'B' denotes base.

| Stage | Output size | PyramidViG-Ti | PyramidViG-S | PyramidViG-M | PyramidViG-B |
|-------|-------------|---------------|--------------|--------------|--------------|
| Stem | $\frac{H}{4} \times \frac{W}{4}$ | Conv×3 | Conv×3 | Conv×3 | Conv×3 |
| Stage 1 | $\frac{H}{4} \times \frac{W}{4}$ | $\begin{bmatrix} D=48 \\ E=4 \\ K=9 \end{bmatrix} \times 2$ | $\begin{bmatrix} D=80 \\ E=4 \\ K=9 \end{bmatrix} \times 2$ | $\begin{bmatrix} D=96 \\ E=4 \\ K=9 \end{bmatrix} \times 2$ | $\begin{bmatrix} D=128 \\ E=4 \\ K=9 \end{bmatrix} \times 2$ |
| Downsample | $\frac{H}{8} \times \frac{W}{8}$ | Conv | Conv | Conv | Conv |
| Stage 2 | $\frac{H}{8} \times \frac{W}{8}$ | $\begin{bmatrix} D=96 \\ E=4 \\ K=9 \end{bmatrix} \times 2$ | $\begin{bmatrix} D=160 \\ E=4 \\ K=9 \end{bmatrix} \times 2$ | $\begin{bmatrix} D=192 \\ E=4 \\ K=9 \end{bmatrix} \times 2$ | $\begin{bmatrix} D=256 \\ E=4 \\ K=9 \end{bmatrix} \times 2$ |
| Downsample | $\frac{H}{16} \times \frac{W}{16}$ | Conv | Conv | Conv | Conv |
| Stage 3 | $\frac{H}{16} \times \frac{W}{16}$ | $\begin{bmatrix} D=240 \\ E=4 \\ K=9 \end{bmatrix} \times 6$ | $\begin{bmatrix} D=400 \\ E=4 \\ K=9 \end{bmatrix} \times 6$ | $\begin{bmatrix} D=384 \\ E=4 \\ K=9 \end{bmatrix} \times 16$ | $\begin{bmatrix} D=512 \\ E=4 \\ K=9 \end{bmatrix} \times 18$ |
| Downsample | $\frac{H}{32} \times \frac{W}{32}$ | Conv | Conv | Conv | Conv |
| Stage 4 | $\frac{H}{32} \times \frac{W}{32}$ | $\begin{bmatrix} D=384 \\ E=4 \\ K=9 \end{bmatrix} \times 2$ | $\begin{bmatrix} D=640 \\ E=4 \\ K=9 \end{bmatrix} \times 2$ | $\begin{bmatrix} D=768 \\ E=4 \\ K=9 \end{bmatrix} \times 2$ | $\begin{bmatrix} D=1024 \\ E=4 \\ K=9 \end{bmatrix} \times 2$ |
| Head | $1 \times 1$ | Pooling & MLP | Pooling & MLP | Pooling & MLP | Pooling & MLP |
| Parameters (M) | | 10.7 | 27.3 | 51.7 | 92.6 |
| FLOPs (B) | | 1.7 | 4.6 | 8.9 | 16.8 |

**Positional encoding.** In order to represent the position information of the nodes, we add a positional encoding vector to each node feature:

$$\mathbf{x}_i \leftarrow \mathbf{x}_i + \mathbf{e}_i, \tag{8}$$

where $\mathbf{e}_i \in \mathbb{R}^D$. The absolute positional encoding as described in Eq. 8 is applied in both iostropic and pyramid architectures. For pyramid ViG, we further include relative positional encoding by

following the advanced designs like Swin Transformer [35]. For node $i$ and $j$, the relative positional distance between them is $\mathbf{e}_i^T \mathbf{e}_j$, which will be added into the feature distance for constructing the graph.

## 4 Experiments

In this section, we conduct experiments to demonstrate the effectiveness of ViG models on visual tasks including image recognition and object detection.

### 4.1 Datasets and Experimental Settings

**Datasets.** In image classification task, the widely-used benchmark ImageNet ILSVRC 2012 [43] is used in the following experiments. ImageNet has 1.2M training images and 50K validation images, which belong to 1000 categories. For the license of ImageNet dataset, please refer to `http://www.image-net.org/download`. For object detection, we use COCO 2017 [34] dataset with 80 object categories. COCO 2017 contains 118K training images and 5K validation images. For the licenses of these datasets, please refer to `https://cocodataset.org/#home`.

**Experimental Settings.** For all the ViG models, we utilize dilated aggregation [30] in Grapher module and set the dilated rate as $\lceil l/4 \rceil$ for the $l$-th layer. GELU [19] is used as the nonlinear activation function in Eq. 6 and 7. For ImageNet classification, we use the commonly-used training strategy proposed in DeiT [51] for fair comparison. The data augmentation includes RandAugment [6], Mixup [73], Cutmix [72], random erasing [74] and repeated augment [20]. The details are shown in Table 3. For COCO detection task, we take RetinaNet [33] and Mask R-CNN [16] as the detection frame-

Table 3: Training hyper-parameters for ImageNet.

| (Pyramid) ViG | Ti | S | M | B |
|---|---|---|---|---|
| Epochs | | 300 | | |
| Optimizer | | AdamW [37] | | |
| Batch size | | 1024 | | |
| Start learning rate (LR) | | 2e-3 | | |
| Learning rate schedule | | Cosine | | |
| Warmup epochs | | 20 | | |
| Weight decay | | 0.05 | | |
| Label smoothing [47] | | 0.1 | | |
| Stochastic path [22] | 0.1 | 0.1 | 0.1 | 0.3 |
| Repeated augment [20] | | ✓ | | |
| RandAugment [6] | | ✓ | | |
| Mixup prob. [73] | | 0.8 | | |
| Cutmix prob. [72] | | 1.0 | | |
| Random erasing prob. [74] | | 0.25 | | |
| Exponential moving average | | 0.99996 | | |

works and use our Pyramid ViG as backbone. All the models are trained on COCO 2017 training set in "1×" schedule and evaluated on validation set. We implement the networks using PyTroch [41] and MindSpore [23] and train all our models on 8 NVIDIA V100 GPUs.

Table 4: Results of ViG and other isotropic networks on ImageNet. ♠ CNN, ■ MLP, ♦ Transformer, ★ GNN.

| Model | Resolution | Params (M) | FLOPs (B) | Top-1 | Top-5 |
|---|---|---|---|---|---|
| ♠ ResMLP-S12 conv3x3 [50] | 224×224 | 16.7 | 3.2 | 77.0 | - |
| ♠ ConvMixer-768/32 [52] | 224×224 | 21.1 | 20.9 | 80.2 | - |
| ♠ ConvMixer-1536/20 [52] | 224×224 | 51.6 | 51.4 | 81.4 | - |
| ♦ ViT-B/16 [9] | 384×384 | 86.4 | 55.5 | 77.9 | - |
| ♦ DeiT-Ti [51] | 224×224 | 5.7 | 1.3 | 72.2 | 91.1 |
| ♦ DeiT-S [51] | 224×224 | 22.1 | 4.6 | 79.8 | 95.0 |
| ♦ DeiT-B [51] | 224×224 | 86.4 | 17.6 | 81.8 | 95.7 |
| ■ ResMLP-S24 [50] | 224×224 | 30 | 6.0 | 79.4 | 94.5 |
| ■ ResMLP-B24 [50] | 224×224 | 116 | 23.0 | 81.0 | 95.0 |
| ■ Mixer-B/16 [49] | 224×224 | 59 | 11.7 | 76.4 | - |
| ★ ViG-Ti (ours) | 224×224 | 7.1 | 1.3 | **73.9** | **92.0** |
| ★ ViG-S (ours) | 224×224 | 22.7 | 4.5 | **80.4** | **95.2** |
| ★ ViG-B (ours) | 224×224 | 86.8 | 17.7 | **82.3** | **95.9** |

## 4.2 Main Results on ImageNet

**Isotropic ViG** The neural network with iostropic architecture keeps the feature size unchanged in its main computational body, which is easy to scale and is friendly for hardware acceleration. This scheme is widely used in transformer models for natural language processing [53]. The recent neural networks in vision also explore it such as ConvMixer [49], ViT [9] and ResMLP [50]. We compare our isotropic ViG with the existing iostropic CNNs [50, 49], transformers [9, 51] and MLPs [50, 49] in Table 4. From the results, ViG performs better than other types of networks. For example, our ViG-Ti achieves 73.9% top-1 accuracy which is 1.7% higher than DeiT-Ti model with similar computational cost.

**Pyramid ViG** The pyramid architecture gradually shrinks the spatial size of feature maps as the network deepens, which can leverage the scale-invariant property of images and produce multi-scale features. The advanced networks usually adopt the pyramid architecture, such as ResNet [17], Swin Transformer [35] and CycleMLP [5]. We compare our Pyramid ViG with those representative pyramid networks in Table 5. Our Pyramid ViG series can outperform or be comparable to the state-of-the-art pyramid networks including CNN, MLP and transformer. This indicates that graph neural network can work well on visual tasks and has the potential to be a basic component in computer vision system.

Table 5: Results of Pyramid ViG and other pyramid networks on ImageNet. ♠ CNN, ■ MLP, ♦ Transformer, ★ GNN.

| Model | Resolution | Params (M) | FLOPs (B) | Top-1 | Top-5 |
|---|---|---|---|---|---|
| ♠ ResNet-18 [17, 59] | 224×224 | 12 | 1.8 | 70.6 | 89.7 |
| ♠ ResNet-50 [17, 59] | 224×224 | 25.6 | 4.1 | 79.8 | 95.0 |
| ♠ ResNet-152 [17, 59] | 224×224 | 60.2 | 11.5 | 81.8 | 95.9 |
| ♠ BoTNet-T3 [46] | 224×224 | 33.5 | 7.3 | 81.7 | - |
| ♠ BoTNet-T3 [46] | 224×224 | 54.7 | 10.9 | 82.8 | - |
| ♠ BoTNet-T3 [46] | 256×256 | 75.1 | 19.3 | 83.5 | - |
| ♦ PVT-Tiny [57] | 224×224 | 13.2 | 1.9 | 75.1 | - |
| ♦ PVT-Small [57] | 224×224 | 24.5 | 3.8 | 79.8 | - |
| ♦ PVT-Medium [57] | 224×224 | 44.2 | 6.7 | 81.2 | - |
| ♦ PVT-Large [57] | 224×224 | 61.4 | 9.8 | 81.7 | - |
| ♦ CvT-13 [60] | 224×224 | 20 | 4.5 | 81.6 | - |
| ♦ CvT-21 [60] | 224×224 | 32 | 7.1 | 82.5 | - |
| ♦ CvT-21 [60] | 384×384 | 32 | 24.9 | 83.3 | - |
| ♦ Swin-T [35] | 224×224 | 29 | 4.5 | 81.3 | 95.5 |
| ♦ Swin-S [35] | 224×224 | 50 | 8.7 | 83.0 | 96.2 |
| ♦ Swin-B [35] | 224×224 | 88 | 15.4 | 83.5 | 96.5 |
| ■ CycleMLP-B2 [5] | 224×224 | 27 | 3.9 | 81.6 | - |
| ■ CycleMLP-B3 [5] | 224×224 | 38 | 6.9 | 82.4 | - |
| ■ CycleMLP-B4 [5] | 224×224 | 52 | 10.1 | 83.0 | - |
| ■ Poolformer-S12 [71] | 224×224 | 12 | 2.0 | 77.2 | 93.5 |
| ■ Poolformer-S36 [71] | 224×224 | 31 | 5.2 | 81.4 | 95.5 |
| ■ Poolformer-M48 [71] | 224×224 | 73 | 11.9 | 82.5 | 96.0 |
| ★ Pyramid ViG-Ti (ours) | 224×224 | 10.7 | 1.7 | **78.2** | **94.2** |
| ★ Pyramid ViG-S (ours) | 224×224 | 27.3 | 4.6 | **82.1** | **96.0** |
| ★ Pyramid ViG-M (ours) | 224×224 | 51.7 | 8.9 | **83.1** | **96.4** |
| ★ Pyramid ViG-B (ours) | 224×224 | 92.6 | 16.8 | **83.7** | **96.5** |

## 4.3 Ablation Study

We conduct ablation study of the proposed method on ImageNet classification task and use the isotropic ViG-Ti as the base architecture.

**Type of graph convolution.** We test the representative variants of graph convolution, including EdgeConv [58], GIN [64], GraphSAGE [13] and Max-Relative GraphConv [30]. From table 6, we can see that the top-1 accuracies of different graph convolutions are better than that of DeiT-Ti,

Table 6: ImageNet results of different types of graph convolution. The basic architecture is ViG-Ti.

| GraphConv | Params (M) | FLOPs (B) | Top-1 |
|---|---|---|---|
| EdgeConv [58] | 7.2 | 2.4 | 74.3 |
| GIN [64] | 7.0 | 1.3 | 72.8 |
| GraphSAGE [13] | 7.3 | 1.6 | 74.0 |
| Max-Relative GraphConv [30] | 7.1 | 1.3 | 73.9 |

indicating the flexibility of ViG architecture. Among them, Max-Relative achieves the best trade-off between FLOPs and accuracy. In rest of the experiments, we use Max-Relative GraphConv by default unless specially stated.

**The effects of modules in ViG.** To make graph neural network adaptive to visual task, we introduce FC layers in Grapher module and utilize FFN block for feature transformation. We evaluate the effects of these modules by ablation study. We change the feature dimension of the compared models to make their FLOPs similar, so as to have a fair comparison. From Table 7, we can see that directly utilizing graph convolution for image classification performs poorly. Adding more feature transformation by introducing FC and FFN consistently increase the accuracy.

Table 7: The effects of modules in ViG on ImageNet.

| GraphConv | FC in Grapher module | FFN module | Params (M) | FLOPs (B) | Top-1 |
|---|---|---|---|---|---|
| ✓ | ✗ | ✗ | 5.8 | 1.4 | 67.0 |
| ✓ | ✓ | ✗ | 4.4 | 1.4 | 73.4 |
| ✓ | ✗ | ✓ | 7.7 | 1.3 | 73.6 |
| ✓ | ✓ | ✓ | 7.1 | 1.3 | 73.9 |

**The number of neighbors.** In the process of constructing graph, the number of neighbor nodes $K$ is a hyperparameter controlling the aggregated range. Too few neighbors will degrade information exchange, while too many neighbors will lead to over-smoothing. We tune $K$ from 3 to 20 and show the results in Table 8. We can see that the number of neighbor nodes in the range from 9 to 15 can perform well on ImageNet classification task.

Table 8: Top-1 accuracy *vs.* $K$ on ImageNet.

| $K$ | 3 | 6 | 9 | 12 | 15 | 20 | 9 to 18 |
|---|---|---|---|---|---|---|---|
| Top-1 | 72.2 | 73.4 | 73.6 | 73.6 | 73.5 | 73.3 | 73.9 |

**The number of heads.** Multi-head update operation allows Grapher module to process node features in different subspaces. The number of heads $h$ in Eq. 5 controls the transformation diversity in subspaces and the FLOPs. We tune $h$ from 1 to 8 and show the results in Table 9. The FLOPs and top-1 accuracy on ImageNet changes slightly for different $h$. We select $h = 4$ as default value for the optimal trade-off between FLOPs and accuracy.

Table 9: Top-1 accuracy *vs.* $h$ on ImageNet.

| $h$ | 1 | 2 | 4 | 6 | 8 |
|---|---|---|---|---|---|
| FLOPs / Top-1 | 1.6B / 74.2 | 1.4B / 74.0 | 1.3B / 73.9 | 1.2B / 73.7 | 1.2B / 73.7 |

### 4.4 Object Detection

We apply our ViG model on object detection task to evaluate its generalization. To have a fair comparison, we utilize the ImageNet pretrained Pyramid ViG-S as the backbone of RetinaNet [33] and Mask R-CNN [16] detection frameworks. The models are trained in the commonly-used "1x" schedule and FLOPs is calculated with 1280×800 input size. From the results in Table 10, we can see that our Pyramid ViG-S performs better than the representative backbones of different types, including ResNet [17], CycleMLP [5] and Swin Transformer [35] on both RetinaNet and Mask R-CNN. The superior results demonstrate the generalization ability of ViG architecture.

Table 10: Object detection and instance segmentation results on COCO val2017. Our Pyramid ViG is compared with other backbones on RetinaNet and Mask R-CNN frameworks.

| Backbone | RetinaNet 1× | | | | | | | |
| --- | --- | --- | --- | --- | --- | --- | --- | --- |
| | Param | FLOPs | mAP | $AP_{50}$ | $AP_{75}$ | $AP_S$ | $AP_M$ | $AP_L$ |
| ResNet50 [17] | 37.7M | 239.3B | 36.3 | 55.3 | 38.6 | 19.3 | 40.0 | 48.8 |
| ResNeXt-101-32x4d [62] | 56.4M | 319B | 39.9 | 59.6 | 42.7 | 22.3 | 44.2 | 52.5 |
| PVT-Small [57] | 34.2M | 226.5B | 40.4 | 61.3 | 44.2 | 25.0 | 42.9 | 55.7 |
| CycleMLP-B2 [5] | 36.5M | 230.9B | 40.6 | 61.4 | 43.2 | 22.9 | 44.4 | 54.5 |
| Swin-T [35] | 38.5M | 244.8B | 41.5 | 62.1 | 44.2 | 25.1 | 44.9 | **55.5** |
| Pyramid ViG-S (ours) | 36.2M | 240.0B | **41.8** | **63.1** | **44.7** | **28.5** | **45.4** | 53.4 |

| Backbone | Mask R-CNN 1× | | | | | | | |
| --- | --- | --- | --- | --- | --- | --- | --- | --- |
| | Param | FLOPs | $AP^b$ | $AP^b_{50}$ | $AP^b_{75}$ | $AP^m$ | $AP^m_{50}$ | $AP^m_{75}$ |
| ResNet50 [17] | 44.2M | 260.1B | 38.0 | 58.6 | 41.4 | 34.4 | 55.1 | 36.7 |
| PVT-Small [57] | 44.1M | 245.1B | 40.4 | 62.9 | 43.8 | 37.8 | 60.1 | 40.3 |
| CycleMLP-B2 [5] | 46.5M | 249.5B | 42.1 | 64.0 | 45.7 | 38.9 | 61.2 | 41.8 |
| PoolFormer-S24 [71] | 41.0M | - | 40.1 | 62.2 | 43.4 | 37.0 | 59.1 | 39.6 |
| Swin-T [35] | 47.8M | 264.0B | 42.2 | 64.6 | **46.2** | 39.1 | 61.6 | **42.0** |
| Pyramid ViG-S (ours) | 45.8M | 258.8B | **42.6** | **65.2** | 46.0 | **39.4** | **62.4** | 41.6 |

## 4.5 Visualization

To better understand how our ViG model works, we visualize the constructed graph structure in ViG-S. In Figure 4, we show the graphs of two samples in different depths (the 1st and the 12th blocks). The pentagram is the center node, and the nodes with the same color are its neighbors. Two center nodes are visualized as drawing all the edges will be messy. We can observe that our model can select the content-related nodes as the first order neighbors. In the shallow layer, the neighbor nodes tend to be selected based on low-level and local features, such as color and texture. In the deep layer, the neighbors of the center nodes are more semantic and belong to the same category. Our ViG network can gradually link the nodes by its content and semantic representation and help to better recognize the objects.

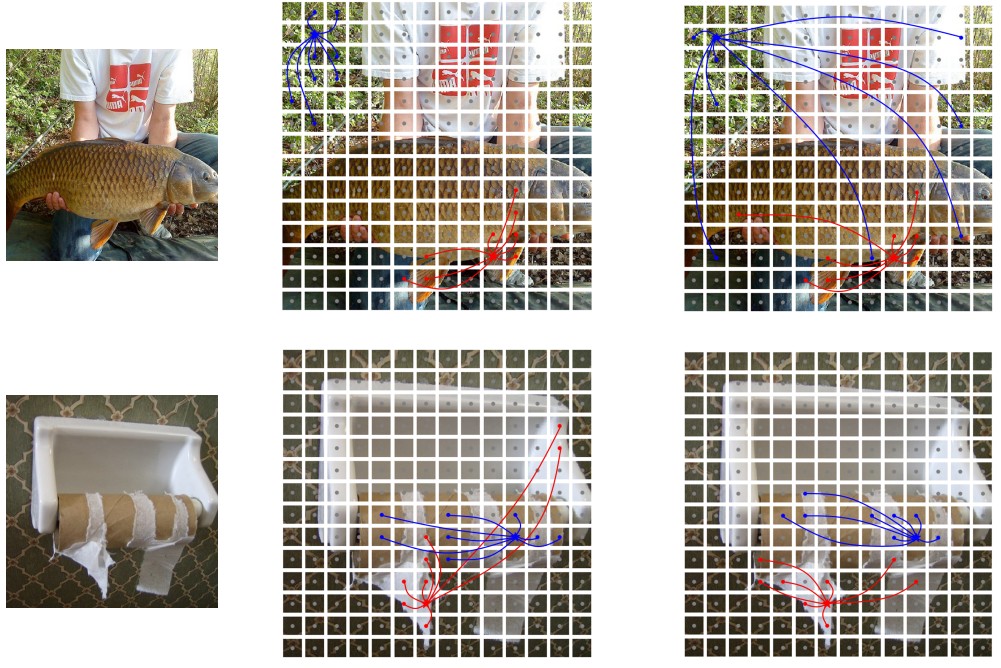

(a) Input image.   (b) Graph connection in the 1st block.   (c) Graph connection in the 12th block.

Figure 4: Visualization of the constructed graph structure. The pentagram is the center node, and the nodes with the same color are its neighbors in the graph.

# 5 Conclusion

In this work, we pioneer to study representing the image as graph data and leverage graph neural network for visual tasks. We divide the image into a number of patches and view them as nodes. Constructing graph based on these nodes can better represent the irregular and complex objects in the wild. Directly using graph convolution on the image graph structure has over-smoothing problem and performs poorly. We introduce more feature transformation inside each node to encourage the information diversity. Based on the graph representation of images and improved graph block, we build our vision GNN (ViG) networks with both isotropic and pyramid architectures. Extensive experiments on image recognition and object detection demonstrate the superiority of the proposed ViG architecture. We hope this pioneering work on vision GNN can serve as a basic architecture for general visual tasks.

## Acknowledgement

This research is supported by NSFC (62072449, 61872345), National Key R&D Program of China (2021YFB1715800), and Macau Science &Tech. Fund (0018/2019/AKP). We gratefully acknowledge the support of MindSpore, CANN (Compute Architecture for Neural Networks) and Ascend AI Processor used for this research.

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
