# Vision GNN: An Image is Worth Graph of Nodes

**Kai Han**[1,2*]    **Yunhe Wang**[2*]    **Jianyuan Guo**[2]    **Yehui Tang**[2,3]    **Enhua Wu**[1,4]
[1]State Key Lab of Computer Science, ISCAS & UCAS
[2]Huawei Noah's Ark Lab
[3]Peking University   [4]University of Macau
{kai.han,yunhe.wang}@huawei.com, weh@ios.ac.cn

## 1   Theoretical Analysis

In our ViG block, we propose to increase feature diversity in nodes by utilizing more feature transformation such as FFN module. We show the empirical comparison between vanilla ResGCN and our ViG model in our paper. Here we make a simple theoretical analysis of the benefit of FFN module in ViG on increasing the feature diversity. Given the output features of graph convolution $X \in \mathbb{R}^{N \times D}$, the feature diversity [?] is measured as

$$\gamma(X) = \|X - \mathbf{1}\mathbf{x}^T\|, \quad \text{where} \quad \mathbf{x} = \arg\min_{\mathbf{x}} \|X - \mathbf{1}\mathbf{x}^T\|, \tag{1}$$

where $\| \cdot \|$ is the $\ell_{1,\infty}$ norm of a matrix. By applying FFN module on the features, we have the following theorem.

**Theorem 1.** *Given a FFN module, the diversity $\gamma(FFN(X))$ of its output features satisfies*

$$\gamma(FFN(X)) \leq \lambda\gamma(X), \tag{2}$$

*where $\lambda$ is the Lipschitz constant of FFN with respect to p-norm for $p \in [1, \infty]$.*

*Proof.* The FFN includes weight matrix multiplication, bias addition and elementwise nonlinear function, which all preserve the constancy-across-rows property of $FFN(\mathbf{1}\mathbf{x}^T)$. Therefore, we have

$$\begin{aligned}
\gamma(\text{FFN}(X)) &= \|\text{FFN}(X) - \mathbf{1}\mathbf{x'}^T\|_p \\
&\leq \|\text{FFN}(X) - \text{FFN}(\mathbf{1}\mathbf{x}^T)\|_p \quad &&\triangleright \text{FFN preserves constancy-across-rows.} \\
&\leq \lambda\|X - \mathbf{1}\mathbf{x}^T\|_p \quad &&\triangleright \text{Definition of Lipschitz constant.} \\
&= \lambda\gamma(X),
\end{aligned}$$

$\square$

The Lipschitz constant of FFN is related to the norm of weight matrices and is usually much larger than 1 [?]. Thus, the Theorem 1 shows that introducing $\gamma(\text{FFN}(X))$ in our ViG block tends to improve the feature diversity in graph neural network.

## 2   Pseudocode

The proposed Vision GNN framework is easy to be implemented based on the commonly-used layers without introducing complex operations. The pseudocode of the core part, *i.e.*, ViG block is shown in Algorithm 1.

---

**Algorithm 1** PyTorch-like Code of ViG Block

---

*Equal contribution.

```python
import torch.nn as nn
from gcn_lib.dense.torch_vertex import DynConv2d
# gcn_lib is downloaded from https://github.com/lightaime/deep_gcns_torch

class GrapherModule(nn.Module):
    """Grapher module with graph conv and FC layers
    """
    def __init__(self, in_channels, hidden_channels, k=9, dilation=1, drop_path=0.0):
        super(GrapherModule, self).__init__()
        self.fc1 = nn.Sequential(
            nn.Conv2d(in_channels, in_channels, 1, stride=1, padding=0),
            nn.BatchNorm2d(in_channels),
        )
        self.graph_conv = nn.Sequential(
            DynConv2d(in_channels, hidden_channels, k, dilation, act=None),
            nn.BatchNorm2d(hidden_channels),
            nn.GELU(),
        )
        self.fc2 = nn.Sequential(
            nn.Conv2d(hidden_channels, in_channels, 1, stride=1, padding=0),
            nn.BatchNorm2d(in_channels),
        )
        self.drop_path = DropPath(drop_path) if drop_path > 0. else nn.Identity()

    def forward(self, x):
        B, C, H, W = x.shape
        x = x.reshape(B, C, -1, 1).contiguous()
        shortcut = x
        x = self.fc1(x)
        x = self.graph_conv(x)
        x = self.fc2(x)
        x = self.drop_path(x) + shortcut
        return x.reshape(B, C, H, W)

class FFNModule(nn.Module):
    """Feed-forward Network
    """
    def __init__(self, in_channels, hidden_channels, drop_path=0.0):
        super(FFNModule, self).__init__()
        self.fc1 = nn.Sequential(
            nn.Conv2d(in_channels, hidden_channels, 1, stride=1, padding=0),
            nn.BatchNorm2d(hidden_channels),
            nn.GELU()
        )
        self.fc2 = nn.Sequential(
            nn.Conv2d(hidden_channels, in_channels, 1, stride=1, padding=0),
            nn.BatchNorm2d(in_channels),
        )
        self.drop_path = DropPath(drop_path) if drop_path > 0. else nn.Identity()

    def forward(self, x):
        shortcut = x
        x = self.fc1(x)
        x = self.fc2(x)
        x = self.drop_path(x) + shortcut
        return x

class ViGBlock(nn.Module):
    """ViG block with Grapher and FFN modules
    """
    def __init__(self, channels, k, dilation, drop_path=0.0):
        super(ViGBlock, self).__init__()
        self.grapher = GrapherModule(channels, channels * 2, k, dilation, drop_path)
        self.ffn = FFNModule(channels, channels * 4, drop_path)

    def forward(self, x):
        x = self.grapher(x)
        x = self.ffn(x)
        return x
```