# OpenReview forum: "Vision GNN: An Image is Worth Graph of Nodes"
_NeurIPS.cc/2022/Conference — NeurIPS 2022 Accept_

### Official Review · Reviewer_CG6g · 2022-07-06

**Rating:** 8
**Confidence:** 4
**Soundness:** 3 good
**Presentation:** 3 good
**Contribution:** 4 excellent

**Summary:**

This paper introduces a GNN-based backbone model for image tasks that is mainly built on the GNN layers and FC layers. Basically, it splits an image into patches and takes each patch as a node to construct the graph structure. A GNN-based architecture is utilized to the graph for visual representation learning. To address the degradation of feature diversity, more transformation of feature is added in the network. Extensive experiments of image classification and object detection show that the proposed Vision GNN can outperform representative convolutional networks and transformers with similar number of parameters.

**Questions:**

1. In the default setting for ImageNet, there are 196 nodes for an image. How does the number of nodes affects the performance of ViG?
2. In the Mask-RCNN experiment, the overall AP of Pyramid ViG is better than Swin but the AP_75. Could you explain this phenomenon?



**Limitations:**

The limitations are addressed.

**Strengths And Weaknesses:**

Strengths:
+ This work is clearly motivated and well written
The background of the research, the motivation of graph representation for images and the related work are all clearly stated and summarized.
+ The first GNN-based backbone for visual tasks
The simple yet effective Vision GNN is introduced by adapting the GNN with reasonable modification by patch-based graph construction and adding more node feature transformations.
+ Extensive evaluation and impressive results
This paper demonstrates the power of the Vision GNN model on ImageNet and COCO datasets, outperforming representative CNN and transformer models. The results are impressive and interesting.

Weaknesses:
- Based on my experience, the inference speed of GNN is not as fast as CNN. It’s a general issue for the researchers to explore for mobile applications of Vision GNN.
- Typos: In line41: GCN->Grapher. In supplemental material, the output channel of self.fc1 in FFN should be `hidden_channels` rather than `in_channels`.

---

> ### Author Response · Authors · 2022-08-02
> **Response to Reviewer CG6g**
>
> Thanks for the valuable comments. We respond to weaknesses and questions in the following.
>
> > **Q1:**
> Based on my experience, the inference speed of GNN is not as fast as CNN. It’s a general issue for the researchers to explore for mobile applications of Vision GNN.
>
> - **A1:**
> With similar parameters and FLOPs, the GPU latency of ViG has no advantage over convolutional network or Transformer. The main parameters and FLOPs are occupied by the fully-connected layers and graph convolution layers which are common operations. The model compression and acceleration of GNN is an important topic for future research.
>
> > **Q2:**
> Typos: In line41: GCN->Grapher. In supplemental material, the output channel of self.fc1 in FFN should be hidden_channels rather than in_channels.
>
> - **A2:**
> Thanks. We'll correct the typos and improve the writing.
>
> > **Q3:**
> In the default setting for ImageNet, there are 196 nodes for an image. How does the number of nodes affects the performance of ViG?
>
> - **A3:**
> We evalute the effect of the number of nodes. The number 196 is the proper one for visual recognition, as larger number leads to more computational cost and smaller number works not so well. Thus, we empirically use 196 nodes for an 224x224 image.
>
> |#nodes|49|196|784|
> |-|-|-|-|
> |FLOPs|0.4G|1.3G|4.8G|
> |Top-1|67.7|73.9|73.2|
>
> > **Q4:**
> In the Mask-RCNN experiment, the overall AP of Pyramid ViG is better than Swin but the AP_75. Could you explain this phenomenon?
>
> - **A4:**
> The overall AP (IoU=.50:.05:.95) of ViG is higher than that of Swin, which denotes that ViG has better detection performance under most of IoU thresholds. The AP_75 of ViG is slightly lower than that of Swin, which denotes the localization ability at 0.75 IoU. Averaging over IoUs rewards detectors with better localization. In COCO competition, AP (IoU=.50:.05:.95) is considered the single most important metric when considering performance on COCO.

---

> > ### Comment · Reviewer_CG6g · 2022-08-09
> > **According to the response**
> >
> > After seeing other reviewers' comments and the received freedback from the authors, I keep my score as is. The paper is clear and has its clear novelty beyond vision transformer.

---

### Official Review · Reviewer_aRD2 · 2022-07-06

**Rating:** 8
**Confidence:** 5
**Soundness:** 3 good
**Presentation:** 3 good
**Contribution:** 4 excellent

**Summary:**

Recently, ConvNets and transformers have achieved state-of-the-art results on various visual recognition tasks. This paper explores graph neural networks (GNN) for visual tasks. By constructing a graph structure from the input image, the paper discusses the differences and advantages of graph structure over grid and sequence structure. GNN is applied on the graph data and FFN is introduced for improving the feature diversity. The obtained ViG backbone can achieve comparable and even better performance than the SOTA ConvNets and transformers.

**Questions:**

See questions in the weaknesses part.

**Limitations:**

The limitations and potential negative societal impact were addressed in the paper.

**Strengths And Weaknesses:**

Strengths

The paper is easy to follow and well-written.
The pioneering exploration of GNN as vision backbone is inspiring to more works. This work will much appeal to the community.
The experiments on image classification and object detection show the effectiveness of vision GNN.

Weaknesses

- In different layers, will the constructed graph structure be updated?
- How ViG is used in object detection is not described in detail. Please include the implementation details.
- Using graph neural networks as a component (although not as the backbone) for image recognition has been investigated in some previous works [1,2]. Please discuss with these works.

[1] Chen, Zhao-Min, et al. "Multi-label image recognition with graph convolutional networks." Proceedings of the IEEE/CVF conference on computer vision and pattern recognition. 2019.
[2] Shen, Yantao, et al. "Person re-identification with deep similarity-guided graph neural network." Proceedings of the European conference on computer vision (ECCV). 2018.

---

> ### Author Response · Authors · 2022-08-02
> **Response to Reviewer aRD2**
>
> Thanks for the valuable comments. We respond to weaknesses and questions in the following.
>
> > **Q1:**
> In different layers, will the constructed graph structure be updated?
>
> - **A1:**
> Yes, the constructed graph structure will be updated in different layers. After aggregation and transformation of nodes, the node features change and the graph edges should also be reconstructed.
>
> > **Q2:**
> How ViG is used in object detection is not described in detail. Please include the implementation details.
>
> - **A2:**
> The ViG based object detection models are implemented using MMDetection [1]. We utilize the ImageNet pretrained Pyramid ViG-S as the backbone of RetinaNet and Mask R-CNN. To process inputs with different size, the position encoding is resized to match different inputs. The output features of all 4 stages are fed into FPN. The models are trained in the commonly-used “1x” scheduler. We'll include the details in the paper.
>
> [1] Chen, Kai, et al. "MMDetection: Open mmlab detection toolbox and benchmark." arXiv preprint arXiv:1906.07155 (2019).
>
> > **Q3:**
> Using graph neural networks as a component (although not as the backbone) for image recognition has been investigated in some previous works [1,2]. Please discuss with these works.
>
> - **A3:**
> ML-GCN [2] utilizes GCN to process label dependency based on the outputs of CNN. It builds a directed graph over the object labels (nodes), and GCN is learned to map this label graph into a set of inter-dependent object classifiers. SGGNN [3] extract features of person images using CNN and utilizes GNN to update the pairwise relationships between probe-gallery pairs. These previous works of GNN for visual tasks mainly utilize GNN as a post component to model relationships between objects, which is much different from the backbone network ViG. We'll include these discussion in the final version.
>
> [2] Chen, Zhao-Min, et al. "Multi-label image recognition with graph convolutional networks." Proceedings of the IEEE/CVF conference on computer vision and pattern recognition. 2019.
>
> [3] Shen, Yantao, et al. "Person re-identification with deep similarity-guided graph neural network." Proceedings of the European conference on computer vision (ECCV). 2018.

---

### Official Review · Reviewer_D9bm · 2022-07-11

**Rating:** 7
**Confidence:** 5
**Soundness:** 3 good
**Presentation:** 3 good
**Contribution:** 3 good

**Summary:**

This manuscript proposes a new kind of backbone named (ViG), which represents the image as a graph and extracts graph-level features for vision tasks. Specifically, the input image is separated into patches as nodes in a graph. Grapher module and FFN module are used to aggregate the information among nodes and transfer feature space. Isotropic and pyramid architectures are proposed to build models of different sizes. The ViG are compared with other SOTA backbones on both image classification task and object detection task.

**Questions:**

- For Tab.8, what’s the meaning of "9 to 18" in the last column?

- The authors compare the ViG with several Transformer-based methods in Tab. 4 in terms of both parameters and FLOPs. What about the real inference time of the proposed method in a standard GPU platform?


**Limitations:**

Yes

**Strengths And Weaknesses:**

Pros:

- Representing an image as graph is novel and interesting.

- The proposed grapher module and FFN are soundness. Also, the visualization of the graph structure indeed shows that the proposed model has learned meaningful relationships among image patches.

- The ablation studies and experiments on several vision tasks are good. Both isotropic and pyramid architectures show the effectiveness of the proposed ViG model.

Cons:

- Some parts of the manuscript are unclear. For example, how to initialize a graph of an image is unclear. It is said the graph is built based on K nearest neighbors, but how to compute the KNN is unclear. Is the KNN constructed based on the position or the similarity of the feature?

- Does the method to compute the KNN influence the performance?

- It would be great if the authors can show more visualization cases of the constructed graph under a more complicated scenario that several objects are involved.

---

> ### Author Response · Authors · 2022-08-01
> **To Reviewer D9bm**
>
> Thanks for the valuable comments. We respond to weaknesses and questions in the following.
>
> >**Q1:** Some parts of the manuscript are unclear. For example, how to initialize a graph of an image is unclear. It is said the graph is built based on K nearest neighbors, but how to compute the KNN is unclear. Is the KNN constructed based on the position or the similarity of the feature?
> - **A1:**
> For the construction of the graph, each node is connected with its K nearest neighbors. The KNN is based on the Euclidean distance between node features. The position information is introduced by the position encoding. We'll improve the presentation of these parts.
>
> >**Q2:** Does the method to compute the KNN influence the performance?
> - **A2:**
> Thanks for the question. We compare different distance metrics between node features including Euclidean distance (our default manner), Manhattan distance and dot product. From the results, we can see that the method to compute the KNN slightly influence the performance.
>
> |KNN metric|Euclidean distance|Manhattan distance|Dot product|
> |-|-|-|-|
> |Top-1|73.9|73.6|73.8|
>
> > **Q3:** It would be great if the authors can show more visualization cases of the constructed graph under a more complicated scenario that several objects are involved.
> - **A3:**
> Thanks for the suggestion. We provide visualization examples for the images with more objects in the anonymous links: xxx and xxx. We can see that give a anchor node, the nodes with the same semantic content will be connected, since the training object is to recognize the image category.
>
> >**Q4:** For Tab.8, what’s the meaning of "9 to 18" in the last column?
> - **A4:**
> Sorry for the unclear notation. The number in the first row of Table 8 means the value of `K` in KNN. "9 to 18" denotes the value of `K` increase from 9 to 18 as layer goes deeper.
>
> >**Q5:** The authors compare the ViG with several Transformer-based methods in Tab. 4 in terms of both parameters and FLOPs. What about the real inference time of the proposed method in a standard GPU platform?
> - **A5:**
> With similar parameters and FLOPs, the GPU latency of ViG has no advantage over convolutional network or Transformer. The main parameters and FLOPs are occupied by the fully-connected layers and graph convolution layers which are common operations. The model compression and acceleration of GNN is an important topic for future research.

---

### Official Review · Reviewer_5KPK · 2022-07-11

**Rating:** 4
**Confidence:** 4
**Soundness:** 3 good
**Presentation:** 3 good
**Contribution:** 2 fair

**Summary:**

In this paper, the authors proposed to represent the image as a graph structure and introduce a graph neural network (ViG) architecture to extract graph level feature for visual tasks. The graph neural network can be aligned with standard vision transformers with many shared micro designs. Images are spilt into patches as nodes in graphs. Each node is connected with its neighborhoods. The ViG network is in a hierarchical feature extraction style like that of Swin Transformer. The authors conducted extensive experiments on image recognition and object detection and comaprable performance with other state-of-the-art methods.

**Questions:**

It seems that when compared with their ViT-B and Swin-B competitor, either ViG-B or Pyramid ViG-B show obvious advantages on ImageNet classification results. I am curious about whether it's the training tricks rather than the algorithm itself that brings the performance gains.

**Ethics Review Area:**

["I don’t know"]

**Limitations:**

I suggest the authors check whether it's possible to use image intrinsic structures proposed in GraphFPN  (GraphFPN: Graph Feature Pyramid Network for Object Detection)  to guide the feature learning of ViG. It will be very interesting then.

**Strengths And Weaknesses:**

***Strength***
1. This paper is well-organized and can be easily understood by readers. The technical details are introduced clearly.
2. The authors conducted extensive experiments on multiple benchmarks to investigate the effectiveness of different modules and designs in this paper.

***Weakness***
1. The idea of graph neural network for visual recognition is appealing, but it seems to be great to expoloits image structures to adjust deep network structures. However, it seems that ViG just takes the simplest setting of graph neural network. It looks like a complex verision of vision transformer. The hierarchical structures involved is in a Swin Transformer style. Since there are a lot of techniques in ViT used, such as multihead attention, feed forward network and etc. I am a little confused about whether ViG is just a simplfied version of Swin Transformer.
2. The feature dimensions. resolutions and other settings listed in Table 2 are very similar to that of Swin Transformer.  So when there are $224 \times 224$ inputs, ViG outputs $7 times 7$ feature maps. But in Table 5, I don't find obvious advantage of ViG when compared with Swin Transformer.
3. Another big problem is that Table 3 lists a lot of tricks used in ViG. According to the MAE paper (Masked Autoencoders Are Scalable Vision Learners, CVPR 2022), a vanilla ViT-B model (86M parameters) can get 82.3% top 1 ImageNet accuracy with the Exponential Moving Average trick (EMA). So the proposed VIG doesn't show any advantages compared with the ViT-B model (in Table 1).

---

> ### Author Response · Authors · 2022-08-02
> **Response to Reviewer 5KPK**
>
> Thanks for the valuable comments. We respond to weaknesses and questions in the following.
>
> > **Q1:** The idea of graph neural network for visual recognition is appealing, but it seems to be great to expoloits image structures to adjust deep network structures. However, it seems that ViG just takes the simplest setting of graph neural network. It looks like a complex verision of vision transformer. The hierarchical structures involved is in a Swin Transformer style. Since there are a lot of techniques in ViT used, such as multihead attention, feed forward network and etc. I am a little confused about whether ViG is just a simplfied version of Swin Transformer.
>
> - **A1:**
> ViG is much different from Swin Transformer:
> 1. To capture spatial information, ViG utilizes graph convolution to aggregating nodes, while Swin Transformer utilizes self-attention among tokens.
> 2. Swin Transformer introduces shifted window for locality inductive bias, while ViG needs less inductive bias.
> 3. Swin Transformer represents the image feature as sequence structure, while ViG construct a graph structure for image feature.
> 4. ViG is a graph neural network, while Swin is a self-attention model.
>
>
> > **Q2:** The feature dimensions, resolutions and other settings listed in Table 2 are very similar to that of Swin Transformer. So when there are  inputs, ViG outputs  feature maps. But in Table 5, I don't find obvious advantage of ViG when compared with Swin Transformer.
>
> - **A2:**
> Our work is a pioneering exploration of graph neural network for general visual recognition. It reveals that GNN can also work well for visual tasks and GNN provides another alternative beyond CNN and Transformer.
> With the similar architecture settings, and without specific designs like shifted window, Pyramid ViG models can be competitive and even better than Swin Transformer (Pyramid ViG-S 82.1% vs. Swin-T 81.3%).
>
>
>
> > **Q3:** Another big problem is that Table 3 lists a lot of tricks used in ViG. According to the MAE paper (Masked Autoencoders Are Scalable Vision Learners, CVPR 2022), a vanilla ViT-B model (86M parameters) can get 82.3% top 1 ImageNet accuracy with the Exponential Moving Average trick (EMA). So the proposed VIG doesn't show any advantages compared with the ViT-B model (in Table 1).
>
> - **A3:** In all the experiments of our paper, we used the same training setting in Swin Transformer. Here we provide the comparison for different supervised training settings in DeiT, Swin and MAE papers. We can see that under the same training setting, ViG-B consistently outperforms ViT-B by about 0.5%.
>
> |Model|training setting|\#parameters|top-1|
> |-|-|-|-|
> |ViT-B|DeiT|86.4M|81.8|
> |ViG-B|DeiT|86.8M|82.4|
> |ViT-B|Swin|86.4M|81.9|
> |ViG-B|Swin|86.8M|82.3|
> |ViT-B|MAE|86.4M|82.3|
> |ViG-B|MAE|86.8M|82.7|
>
>
> > **Q4:** It seems that when compared with their ViT-B and Swin-B competitor, either ViG-B or Pyramid ViG-B show obvious advantages on ImageNet classification results. I am curious about whether it's the training tricks rather than the algorithm itself that brings the performance gains.
>
> - **A4:** For ViT-B vs. ViG-B, we compare them in different training settings fairly in the above table. For Swin-B vs. Pyramid ViG-B, their training settings are extactly the same, so the comparison is fair. These experimental results show that it's the ViG itself that brings the performance gains.
>
> > **Q5:** I suggest the authors check whether it's possible to use image intrinsic structures proposed in GraphFPN (GraphFPN: Graph Feature Pyramid Network for Object Detection) to guide the feature learning of ViG. It will be very interesting then.
>
> - **A5:**
> Without complex design, ViG simply using uniform division of image can obtain a competitive performance.
>
> GraphFPN is a "CNN backbone + GNN head" network for object detection built on a superpixel hierarchy. It's a good proposal to use image intrinsic structures to guide the feature learning of ViG. Nevertheless, there are still several problems to overcome:
>
> 1. For each input image, the COB algorithm is applied to obtain a hierarchical segmentation. The superpixel segmentation algorithm including COB requires a large latency which will be a burden for ViG.
> 2. For training, the obtained superpixels have various size. How to transform different-size superpixels to same-size vectors as inputs of ViG is a open problem.
> 3. Adding a segmentation process before ViG will corrupt the end-to-end training manner.
>
> These topics will be good directions for future research.

---

### Meta-Review · Area_Chair_pU4c · 2022-08-25

**Recommendation:** Accept
**Confidence:** Certain

**Metareview:**

This paper proposes to explore the graph structure of images by considering patches as nodes, where the graph is constructed by connecting nearest neighbors. Extensive experiments on various visual tasks, i.e., image recognition and object detection have demonstrated the effectiveness of the proposed ViG. All the reviewers agree on the inspiring and promising exploration. The paper is also well-written and the experimental results are impressive.

**Award:**

No

---

### Decision · Program_Chairs · 2022-09-14

Accept